# A Comprehensive Review of the Potential Role of Liquid Biopsy as a Diagnostic, Prognostic, and Predictive Biomarker in Pancreatic Ductal Adenocarcinoma

**DOI:** 10.3390/cells13010003

**Published:** 2023-12-19

**Authors:** Kosta Stosic, Oier Azurmendi Senar, Jawad Tarfouss, Christelle Bouchart, Julie Navez, Jean-Luc Van Laethem, Tatjana Arsenijevic

**Affiliations:** 1Laboratory of Experimental Gastroenterology, Université Libre de Bruxelles, 1070 Brussels, Belgiumoier.azurmendi@ulb.be (O.A.S.); christelle.bouchart@hubruxelles.be (C.B.);; 2Department of Radiation Oncology, Hopital Universitaire de Bruxelles (H.U.B.), Université Libre de Bruxelles (ULB), 1070 Brussels, Belgium; 3Department of Hepato-Biliary-Pancreatic Surgery, Hopital Universitaire de Bruxelles (H.U.B.), Université Libre de Bruxelles (ULB), 1070 Brussels, Belgium; 4Department of Gastroenterology, Hepatology and Digestive Oncology, Hopital Universitaire de Bruxelles (H.U.B.), Université Libre de Bruxelles (ULB), Route de Lennik 808, 1070 Brussels, Belgium

**Keywords:** pancreatic cancer, tumoural heterogeneity, liquid biopsy

## Abstract

Pancreatic ductal adenocarcinoma is one of the most lethal malignant diseases, with a mortality rate being close to incidence. Due to its heterogeneity and plasticity, as well as the lack of distinct symptoms in the early phases, it is very often diagnosed at an advanced stage, resulting in poor prognosis. Traditional tissue biopsies remain the gold standard for making a diagnosis, but have an obvious disadvantage in their inapplicability for frequent sampling. Blood-based biopsies represent a non-invasive method which potentially offers easy and repeated sampling, leading to the early detection and real-time monitoring of the disease and hopefully an accurate prognosis. Given the urgent need for a reliable biomarker that can estimate a patient’s condition and response to an assigned treatment, blood-based biopsies are emerging as a potential new tool for improving patients’ survival and surveillance. In this article, we discuss the current advances and challenges in using liquid biopsies for pancreatic cancer, focusing on circulating tumour DNA (ctDNA), extracellular vesicles (EVs), and circulating tumour cells (CTCs), and compare the performance and reliability of different biomarkers and combinations of biomarkers.

## 1. Introduction

Pancreatic ductal adenocarcinoma (PDAC) is one of the deadliest malignant diseases, with a five-year survival rate of 12% [1]. Not only does PDAC not decline, it is predicted to become the second leading cause of cancer-related death in the Western world within less than a decade [2,3]. There are several reasons for such a dismal prognosis, including the absence of distinct symptoms at the early stages of the disease, a lack of reliable screening markers, and a high metastasis rate. 

The only currently available curative option is surgical resection followed by adjuvant chemotherapy. Due to the abovementioned reasons, only one third of the patients with PDAC are diagnosed with a non-metastatic disease, for whom a project of surgery could be considered [4,5]. Nowadays two main chemotherapeutic regimens have proven their efficacy in PDAC management: FOLFIRINOX and nab-paclitaxel/gemcitabine in metastatic disease and the former in an adjuvant setting after surgery [6,7]. These regimens are now increasingly used in a neoadjuvant setting to downstage PDAC that cannot be curatively resected immediately [8]. 

PDAC biology is extremely heterogenous and complex, and its heterogeneity is considered one of the main reasons for its resistance to treatment. Tumour heterogeneity exists not only among patients (intertumoural heterogeneity), but within the same tumour (intratumoural heterogeneity) [9]. PDAC is characterised by genomic instability and a high mutation rate, with undruggable Kirsten rat sarcoma virus gene (KRAS) mutations detected in more than 90% of PDAC cases [10]. In addition, the PDAC tumoural microenvironment (TME) is highly complex and defined by extensive desmoplastic stroma, being an additional source of intratumoural and spatial heterogeneity.

Finally, temporal heterogeneity originates from PDAC change over the course of time and treatment [9].

Endoscopically or transcutaneously US-guided tissue sampling, the most widely used methods for PDAC diagnosis, are invasive surgical approaches that provide information limited to a single point in space and might therefore fail to capture the complex PDAC tumoural biology and heterogeneity [11,12,13,14,15,16,17]. Furthermore, due to the unfavorable location of the pancreas and the invasiveness of the procedure, repeated tissue sampling for monitoring analysis is difficult [18,19]. Human samples that could serve as a source for a biomarker disclosure should be easily accessible, not too complex for isolation, and to be able to showcase changes in different pathological stages. Here, liquid biopsies are emerging as a valuable tool to monitor PDAC heterogeneity [20,21,22,23,24,25]. So far, three of the most prominent cancer-related blood biopsy components have been cell-free DNA (cfDNA), exosomes, and circulating tumour cells (CTCs). The research on cfDNA in PDAC dates back to the 1980s, when Shapiro et al. found a significant increase in cfDNA levels in about 90% of PDAC patients [26]. Since then, KRAS mutations are most extensively researched circulating tumour DNA (ctDNA) biomarkers in PDAC. Nevertheless, in the pooled analysis, a KRAS mutation was detected in 70–80% of the individuals with locally advanced or metastatic PDAC and 30–68% of those with resectable tumours [27]. This may be explained by the fact that the quantity of ctDNA may depend on the number of PDAC cells in the bloodstream, the stage, and the bulk of metastasis. Even though exosomes were also originally described in the early 1980s, interest in this type of extracellular vesicle and its potential use in the fight against cancer has significantly increased in the last 15 years, since the discovery that they harbor mRNA and microRNA. As for the CTCs, although they were first reported in 1869 by Thomas Ashworth, Allard et al. were the first ones who attempted to capture them in patients suffering from PDAC [28].

In this article, we review the advancement in biomarker identification in blood biopsy samples of PDAC patients, focusing on the latest discoveries and advancements in the use of liquid biopsies for the early detection of PDAC, the assessment of prognosis, and the capacity to predict response to therapies.

## 2. Protein Biomarkers Currently in Use in Clinical Settings

The well-known carbohydrate antigen 19-9 (CA19-9) and carcinoembryonic antigen (CEA) are currently the only clinically used blood-based markers for PDAC [29]. A large meta-analysis that included 57 studies representing 3285 patients with pancreatic cancer and 37 studies representing 1882 cases with benign pancreatic disease confirmed an 80% sensitivity and specificity for CA 19-9 (with a 37 U/mL cutoff) and a sensitivity of 45% and specificity of 85% for CEA (5 ng/mL cutoff) [30]. CA 19-9 can be elevated in other malignancies and benign hepato-pancreaticobiliary conditions, contributing to lower diagnostic accuracy for PDAC [31]. Neither of these markers has the necessary accuracy to screen asymptomatic populations, and they are currently used in clinical context in combination with imaging modalities to guide diagnostic and treatment decisions. Furthermore, in 5–10% of PDAC patients CA19-9 is not detectable due to a lack of fucosyltransferase activity caused by homozygous mutations of the FUT3 gene [30]. The identification of novel biomarkers with better performance for PDAC diagnosis and monitoring is urgently needed.

## 3. Cell-Free Nucleic Acids 

### 3.1. Cell-Free DNA

CfDNAs are fragments of DNA that enter the bloodstream through secretion, apoptosis, necrosis, and autophagy [32,33]. The majority of cfDNA molecules have a length that corresponds to the DNA wrapped around a nucleosome [34,35,36]. The fraction of cfDNAs that originates from tumoural cells is called ctDNA (Figure 1). The amount of ctDNA in the blood can vary from less than 0.01% to about 93% and is tightly correlated to the tumour burden [34,37]. ctDNAs have emerged as promising biomarkers for cancer diagnosis and monitoring, as they reflect an elevated level of metabolic activity, necrosis, and apoptosis in the microenvironment of the tumour [38]. ctDNAs can be secreted from primary and metastatic tumours as well as from circulating tumour cells (CTCs) [39]. Unlike CTCs, which are very rare, ctDNAs are detected at higher concentrations in blood biopsies originating from cancer patients, and they are composed of both coding and non-coding genomic and mitochondrial DNA [40]. After cfDNA’s isolation from blood samples, it is necessary to perform mutational analysis to identify ctDNAs, since an increase in total cfDNA levels may be indicative of a non-malignant condition, such as inflammation. Furthermore, the share of ctDNAs in the pool of cfDNAs can differ significantly among patients, which might affect the accuracy of estimations of the tumour size and stage [41].

### 3.2. Isolation of Cell-Free DNA and Circulating Tumour DNA Detection Methods 

Different methods have been developed to analyse ctDNA, each with their own advantages and limitations. One of the earliest techniques used was polymerase chain reaction (PCR) which can amplify specific DNA sequences with high sensitivity and low cost; however, it requires prior knowledge of the mutations to be detected [42]. Later, other more advanced methods were developed, such as digital PCR (dPCR), real-time (quantitative) PCR (qPCR), digital droplet PCR (ddPCR) [43,44,45], and beads–emulsion–amplification–magnetics (BEAMing) [46]. Their advantage is that they can quantify ctDNA with even higher sensitivity (ranging from 0.1% to 0.001%) and high speed, although they still rely on a known mutational profile of the tumour [47]. 

A method that overcomes this limitation is next generation sequencing (NGS), which offers the possibility to analyse a large number of loci with both high sensitivity and high depth, while having the potential to unravel both known and unknown mutations. NGS has the advantage of providing a comprehensive view of the genomic alterations in ctDNA, including single nucleotide variants, structural variations, and copy number variations. However, NGS has lower sensitivity (around 1%) and requires a significant amount of cfDNA, which may not be available from plasma samples [44,48,49]. Several NGS-based methods have been applied to ctDNA analyses, such as tagged amplicon deep sequencing (TAm-Seq) [50], whole genome sequencing (WGS) [51], ion torrent next generation sequencing (Ion-AmpliSeq) [52], cancer personalised profiling by deep sequencing (CAPP-Seq) [53], and sensitive mutation detection using sequencing (SiMSen-Seq) [54], which have improved the sensitivity and specificity of ctDNA detection, offering similar sensitivity for ctDNA detection as that of dPCR [55]. 

### 3.3. Cell-Free RNA

Similarly to DNA, RNAs can also be shed into the bloodstream by cells. For longer RNA molecules, studies are confirming the existence of messenger RNA (mRNA), long non-coding RNA (lncRNA), and circular RNA (circRNA) in blood samples [56]. However, cell-free RNAs (cfRNAs) are usually partially degraded and present in low concentrations, unlike RNA molecules enclosed in exosomes (Section 4.1 and Section 6.3). Tumour-derived mRNAs in circulation and their use in PDAC blood biopsies have been investigated and reported in only a few studies so far [57,58,59], while most RNA-based PDAC diagnoses were performed by measuring the levels of representation of non-coding RNAs (ncRNAs), especially micro RNAs (miRNAs). The obvious drawback of this kind of approach is the uncertainty of the source of the abovementioned RNAs. Altered levels of miRNA a may be a result of existing PDAC, but they might also indicate some other pathological condition or they might simply derive from hematopoietic cells or the immune system [60,61]. 

Many publications attempt to use either newly discovered or already known miRNAs, both alone and as a part of panels of biomarkers [62,63,64]. In some cases, the mentioned panels included both different miRNAs and CA19-9 or CEA, whereas in others they might only consist of miRNA panels. Although some reports claim a sensitivity of nearly 100%, these potential biomarkers lack significant specificity [65]. 

### 3.4. Cell-Free RNA Detection Methods 

Reverse transcription quantitative PCR (RT-qPCR) and ddPCR are the most common methods for the detection and profiling of cfRNAs. RT-qPCR is fast, inexpensive, and has high specificity; however, it has low sensitivity in samples with low cfRNA levels. ddPCR, on the other hand, has high sensitivity and accuracy, as well as greater reproducibility and a lower necessary sample volume (compared to RT-qPCR), but requires tedious assay optimization [66].

## 4. Extracellular Vesicles

Extracellular vesicles (EVs) are another component of liquid biopsies and an emerging tool for blood-based profiling of cancer, including PDAC. Even though there are many types of EVs, the majority of studies so far has focused on exosomes, which originate from the inward budding of the endosomal compartment and have a diameter of 30–150 nm [67,68].

EVs carry diverse types of cargo, including DNA, RNA, proteins, and lipids. The membrane of EVs acts as a protective barrier from nucleases, enabling the existence of larger fragments of nucleic acids than ctDNAs, which are mainly 160–170 bp long [69]. Exosomes are secreted by all cells, tumoural and non-tumoural [70,71]. Therefore, a challenge of using EVs, as well as other constituents of blood biopsies, is that genetic information will be largely diluted with non-cancer cell-derived vesicles, cfDNA, and cfRNA. However, contrary to ctDNA, exosomes can be released by living cells, so they can be present in the bloodstream before necrosis-originating cfNAs, making them useful for the early diagnosis of tumours. In TME, exosomes mediate communication not only among cancer cells (for example to confer chemoresistance), but also between cancer cells and other cell types (such as cancer-associated fibroblasts, PSCs, and β-cells) [72,73,74,75,76,77]. Tumour-derived exosomes can affect biological processes such as immune system suppression, lipolysis induction, diabetes onset, and premetastatic niche formation [78,79]. Some studies suggest that exosomal cargo determines the preferred site of metastasis [80,81,82]. The mRNA expression in exosomes from different PDAC cell lines has been shown to resemble the mRNA signature in the cells of origin [83,84]. However, there have been reports finding differences in the RNA and protein profiles of exosomes and their source cells [85,86,87], suggesting the possibility of an active and selective loading mechanism employed by extracellular vesicles.

### 4.1. Exosome Isolation and Enrichment Methods

Methods for isolating exosomes can be size-based, density-based, or affinity-based. Differential ultracentrifugation (DC) is the most iteratively used technique, followed by several others, such as density gradient ultracentrifugation (DGC), ultrafiltration (UF), precipitation, size-exclusion chromatography (SEC), and immunoaffinity capture [88]. 

These conventional methods are widely available, but they have disadvantages resulting in different characteristics of exosomes in terms of size, surface markers, and contaminants. Immunoaffinity capture has some obvious advantages such as high purity, the ability to separate exosomal sub-populations, and the availability of commercial kits, but it has some potential drawbacks that should not be overlooked, such as exosome damage during elution, costly reagents, low capacity, and finally, the potential that it may yield a mixture of apoptotic bodies and microvesicles [89]. 

Both density and size-based methods result in a contaminated output with a low concentration and specificity, whereas affinity-based techniques offer isolation performances with higher specificity and purity but with low sample yields [47]. Lately, microfluidic devices, such as physical property-based methods, immune-chip capture, and comprehensive separation have been developed in order to boost the performance of exosome isolation [90,91]. They require a smaller sample volume, less time, and lower quantities of reagents and have higher portability compared to conventional methods. Disadvantages include the lack of standardized protocols and the need for further improvements in purity and further cost reductions. In 2020, an efficient and automated platform that included microfluidic chips and a combination of the cutting-edge microfluidic approach and the traditional immunomagnetic bead-based technique was developed. This method offers the possibility to extract a specified subtype of exosomes with a particular protein biomarker [92]. 

## 5. Circulating Tumour Cells (CTCs)

CTCs are a heterogeneous cell population including both viable and apoptotic tumoural cells, originating from the primary tumour or its metastases (Figure 2). They can detach from the tumour and enter the blood or lymph vessels [93]. The number of CTCs in the blood varies widely, from 10~100 per 10^6^–10^8^ white blood cells (depending on how they are enriched) [94], to as low as one cell per 10^8^ [70] or even 10^9^ blood cells [40]. CTCs can be isolated as single cells or as clusters called circulating tumour microemboli (CTM) [38,46,95,96]. CTM may contain different cell types, such as cancer-associated fibroblasts, immune cells, platelets, and pancreatic stellate cells [97,98] and have a higher migration potential and rate of survival than CTCs [99]. 

The most probable reason behind the rareness of CTCs in circulation is their size. They are often three to five times bigger than the size of the capillary openings prior to the entrance into central circulation through the portal vein [100]. Henceforth, the number of CTCs may vary both spatially and temporally, raising the questions of whether the sampling location should be redirected in hope of an early diagnosis and whether CTCs are the real source of systemic metastasis [101,102,103]. Another hypothesis, that does not exclude the first one, concerns immune cell attacks, anoikis, and shear flow as the causes of the low share of CTCs in circulation [104]. 

Cancer cells initially keep the features of the epithelial cells they originate from. Gradually they change and develop different phenotypes in order to avoid apoptosis and create metastatic lesions. The main process in cancer onset and progression is epithelial-to-mesenchymal Transition (EMT), and it represents the main weapon that CTCs use to upgrade their migration ability and invasion capacity [105,106]. Molecules expressed on CTCs, such as the epithelial cell adhesion molecule (EpCAM) and E-cadherin, may change in the course of EMT actions. Molecules that are repeatedly used as CTC reporters are creatine kinase family members [55]. 

CTCs in circulation might be detached from various regions of the same tumour, from several loci, or even from both the primary mass and occult metastasis. Therefore, they may reflect a whole-body image regarding the tumour burden better than biopsies taken from a single spot [94]. Furthermore, unlike ctDNA molecules, CTCs provide not only genetic information, but the RNA and protein signature which is cloaked by the membrane [107,108]. 

Even though PDAC and its CTCs are extremely heterogenous, the mutational profile detected in PDAC-derived CTCs often aligns with those found in the primary tumour and metastatic lymph nodes [109,110]. Kulemann et al. analysed molecular alterations in the CTCs of PDAC patients, mostly focusing on the KRAS mutation status. In contrast to what was expected, they observed significantly better survival (19.4 vs. 7.4 months) in patients with the CTC KRAS mutation in comparison to those who had a wild type KRAS [111]. Two years later, the same group conducted an extended investigation and showed worse median overall survival (OS) in patients with >3 CTCs/mL compared to patients who had values below that cutoff. Surprisingly, results also showed discordance between KRAS alterations in CTCs and corresponding primary tumours, thus emphasizing the heterogeneity in KRAS mutations in CTCs and originating tissue. Interestingly, it was observed that individuals with a CTC KRAS^G12V^ mutation had a significantly longer median OS than those who harbored other mutations or undetectable KRAS changes [112]. Another study explored the potential and use of pharmacogenomic (PGx) modeling of circulating tumour and invasive cells (CTICs) to predict patients’ responses to therapy, progression, and potential resistance according to genetic mutations. The authors were able to stratify patients as “sensitive” and “resistant” to several chemotherapy regimens commonly used in PDAC. Furthermore, they showed that the group labeled as “sensitive” had longer DFS and OS rates than the “resistant” one [113]. 

### Circulating Tumour Cells Isolation and Enrichment Methods 

Different techniques for CTC isolation have been developed over the years, and each of them can be clustered into a particular type of CTC extraction methodology group. the first explorations included reverse transcription-PCR (RT-PCR) and ultra/standard density centrifugation, and were implemented for the isolation of PDAC-derived CTCs (from whole blood) based on the epithelial cell adhesion molecule (EpCAM), CEA, and cytokeratin-20 (CK-20) [114,115,116]. They had poor specificity, exemplified by unsatisfactory detection rates, such as 25% for EpCAM [116], 26% for CEA mRNA [117], and 34% for CK-20 [118]. 

The most cited method for CTC isolation is the CellSearch^®^ (Menarini Silicon Biosystems Inc, Huntington Valley, PA, USA) method, which uses magnetic beads coated with anti-EpCAM and anti-cytokeratin antibodies (for the enrichment of the epithelial population of CTCs) and anti-CD45 (intended for white blood cell depletion). CellSearch^®^ is the only technology approved by the U. S. Food and Drug Administration (FDA) and is currently labeled as the gold standard for CTC isolation in metastatic breast, prostate, and colorectal cancer [70]. Several following studies using CellSearch^®^ reported variable detection rates from 10 to 50% [102,119,120,121,122]. This method enabled diagnoses for 11–48% of patients in a PDAC cohort with at least 53% of patients having either locally advanced or metastatic disease [122]. In contrast, for resectable patients, the detection rate dropped below 7% [123]. Notably, given that CellSearch^®^ technology is used according to the idea that CTCs do not express the CD45 antigen (a feature of the white blood cells), it disregards the fact that CTCs can affix to immune cells and platelets, and hence register as CD45 positive [93]. 

The isolation by size of tumour cells (ISET^®^) (Rarecells DIAGNOSTICS, Paris, France) [124] method may emerge as a potential alternative, as it enables separation according to size and thus being unbiased towards marker molecules. 

The advantage that methods based on size differences hold over antigen-affinity technology is the ability to separate phenotypes independently of marker expression on the cell surface. Nevertheless, all these approaches need to be accompanied by immuno-visualization techniques. Microfluidic methodologies such as CTC-iCHIP [125] and NanoVelcro [126] are capable of separating CTCs according to both antigen affinity and size, thus elevating purity and specificity levels. 

## 6. Diagnostic Potential of PDAC Liquid Biopsy Biomarkers

The ideal situation for clearly delineating between the PDAC and healthy liquid biopsy samples would be the discovery of a robust and unique biomarker with sufficiently high values for both sensitivity (SN) and specificity (SP). Nevertheless, taking into account the heterogeneity of PDAC, as well as the interpatient variability, more and more studies are reorienting towards trying to identify a specific panel combining several potential biomarkers [56]. It has been proposed that a sufficient biomarker or a panel should perform with a minimum of 88% SN and 85% SP in order to be considered valuable [65]. In this paper, we review some of the discoveries that have achieved or surpassed these values. 

### 6.1. Protein Biomarkers 

As already mentioned, CA 19-9 and CEA are the most commonly used biomarkers for the diagnosis of patients with PDAC. However, due to their specificity and sensitivity limitations, the quest for new circulating protein biomarkers with high specificity and sensitivity for PDAC is ongoing. The role of glypican-1 (GPC-1), a heparan sulfate proteoglycan (HSP), in pancreatic cancer diagnosis has been controversial. While Kleeff et al. (1998) reported that GPC-1 was highly expressed in both cancer cells and fibroblasts in human pancreatic cancer samples [127], Melo et al. (2015) found that GPC-1 was enriched on exosomes derived from tumour cells and could be detected in serum with a staggering 100% SN and SP [25] (Table 1). However, these findings were not replicated by subsequent studies. Lai et al. (2017) showed that a panel of exosomal miRNAs (miR-10b, miR-21, miR-30c, miR-181a, and miR-let7a) had better diagnostic performance than GPC-1 for distinguishing PDAC from healthy and chronic pancreatitis (CP) samples [63]. Frampton et al. (2018) also observed no significant difference in GPC-1 levels between PDAC and benign pancreatic disease or healthy pancreas [128]. 

According to several other studies, the evidence for GPC-1 is weak and inconsistent [178,179]. However, two studies have found some positive results with GPC-1 in combination with CD63 [180] or after regional intra-arterial chemotherapy [181]. These studies have limitations such as a low specificity, small sample size, and lack of control groups. Therefore, the role of GPC-1 as a biomarker for PDAC remains unclear and needs further investigation.

Several studies have investigated the performance of different protein biomarkers for detecting PDAC. For example, Li et al. reported that both serum and tissue samples of PDAC patients had higher levels of REG1A and REG1B proteins, with an SN of 92% and SP of 95% (*n* = 44) [131]. However, these findings need to be validated in a larger and independent cohort before they can be applied clinically. In the same year, the diagnostic potential of PIM-1 was also evaluated. Preoperative plasma samples were collected for CP (*n* = 19), pancreatic neuroendocrine tumours PNETs (*n* = 20), other pancreatic tumours (*n* = 29), and healthy individuals (*n* = 20) [132]. The diagnostic performance was measured through receiver operating characteristic (ROC) curve analyses. PIM-1 showed a high SN of 95.6% and a perfect SP of 100% when distinguishing PDAC from healthy volunteers, with an AUC of 0.984. In comparison, CA19.9 had the same SP but a lower SN of 74.4% and an AUC of 0.879. PIM-1 also outperformed CA19.9 when differentiating PDAC from CP; however, it was less effective when compared to other pancreatic tumours and performed poorly in discriminating PDAC from PNET [132]. The large meta-analysis including data from 28 primary studies (with 6127 individuals) was analysed to evaluate the diagnostic performances of various proteins. The study included 6127 participants, including 2770 PDAC patients, 2082 healthy controls, and 1275 samples from benign disease cases. MIC-1 showed the highest accuracy in distinguishing PDAC and healthy samples with an AUC = 0.93, surpassing CA19.9. However, MIC-1 was not as effective as CA19.9 in discriminating PDAC patients from benign disease cases. Adding MIC-1 to CA19.9 did not significantly improve its diagnostic potential [133]. 

Numerous other studies identified relevant single protein markers and multi-protein panels, and the most prominent findings regarding both groups are summarised in Table 1. 

### 6.2. Cell-Free DNA

The most common methods used for cfDNA analysis are mutation detection (mainly KRAS for PDAC) [182] and the determination of the cfDNA quantity [153] and the cfDNA methylation status [154,155,156]. MutKRAS-ctDNA was tested on EUS-FNA as a potential biomarker, compared to CTCs, and combined with CA19.9 but failed to reach the high thresholds [183]. Considering the diagnostic power of cfDNA’s quantity, the highest accuracy was achieved using a cutoff value of 0.208 ng/µL for IPMN vs. HC, 0.2875 ng/µL for PDAC vs. HC, and 0.3590 ng/µL for PDAC vs. IPMN. Corresponding SNs and SPs were 80.95% and 84.21% (for IPMN vs. HC), 83.3% and 92.11% (for PDAC vs. HC), and 75% and 71.43% (for PDAC vs. IPMN) [153]. Even though the use of nanoparticle-enabled MOB technology unveiled an increased frequency of DNA methylation in the genes BNC1 and ADAMTS1, their diagnostic capacity has not lived up to the potential (79% and 89% for BNC1 and 48% and 92% for ADAMTS1). However, when combined into a two-marker panel, the overall SN and SP reached 81% and 85% [155]. In a more recent publication, Eissa et al. tested this two-marker panel on a larger cohort including 92 HC and recorded 97.3% and 91.6% (SN and SP) [156]. That same year, Manoochehri et al. reported the diagnostic performance of hypermethylation of the SST gene in discerning PDAC from HC (SN = 93.3%, SP = 88.9% and AUC of 0.89) [154]. 

### 6.3. Circulating and Exosomal RNAs

As mentioned earlier, RNAs are one of the biomolecules that could serve as a single biomarker besides proteins. Among the various types of RNAs, miRNAs have been the most widely studied and reported. In particular, exosomal RNAs (exoRNAs) have attracted increasing interest as exosomes protect RNAs from degradation by RNases, and exoRNAs derived from tumours reflect the transcriptomic profile of the primary cancer cells more closely. Moreover, exoRNAs have higher abundance than free circulating RNAs. miR-18a expression was significantly increased in tumour tissue (*p* = 0.012) and PDAC cell lines (*p* = 0.015) compared to normal tissue and fibroblasts, as well as in plasma samples of PDAC patients versus controls (*p* < 0.0001), with an AUC value of 0.9369 [160]. Another small study (35 PDAC samples and 15 controls) demonstrated the excellent performance of miR-22-3p (SN of 97.14% and SP of 93.33%, with an AUC value of 0.943), miR-642b-3p (100%, 100%, and AUC = 1.0), and miR-885-5p (100%, 100%, and AUC = 1.0), surpassing the accuracy of CA19.9 (91.43%, 100%, AUC = 0.924) [161]. However, the sample size was small and there was no validation cohort to confirm the findings. A recent study explored the diagnostic potential of ADAM8-positive EVs and related exo-miR-451 and exo-miR-720, as these miRNAs showed significant changes in PDAC vs. healthy samples. The AUC values of these miRNAs were 0.93 and 1, respectively [163]. 

The idea of discovering a single biomarker that could accurately diagnose PDAC is becoming less and less realistic; thus, many studies have focused on developing panels of multiple molecules (mostly consisting of miRNAs or proteins) that can improve the sensitivity and specificity of PDAC detection. These panels often include the measurement and comparison of expression levels of various candidates, followed by the selection of the best ones and the integration with CA19.9 or, less often, CEA. 

To compare the expression levels of seven predetermined miRNAs (miR-16, 21, 155, 181a, 181b, 196a, and 210), the RNAs were isolated from the plasma of 140 PDAC patients, 111 CPs, and 68 healthy controls. All of tested miRNAs seemed to be upregulated when comparing PDAC with CP and PDAC with healthy samples. Logistic modelling has shown that miR-16 and miR-196a were the only ones with the potential to separate patients as single markers. Therefore, by creating a signature that consisted of miR-16, miR-196a, and CA19.9, the distinguishing potential reached an SN of 92% and an SP of 95.6% for PDAC vs. controls (AUC = 0.979) and an SN of 88.4% and an SP of 96.3% for PDAC vs. CP (AUC = 0.956) [166]. Another signature that included CA19.9 and two miRNAs (miR-1290 and miR-1246) was established in a cohort that consisted of 120 PDAC, 40 non-PDAC controls (10 CPs, 15 IPMNs, and 15 PNETs), and 40 healthy control (HC) samples. For PDAC vs. healthy controls, this signature reached an SN of 96.7% and an SP of 97.5%. For PDAC vs. non-PDAC controls, the signature achieved an SN of 92.5% and an SP of 90% [167]. 

Several years ago, a diagnostic panel containing eight exo-mRNAs (CLDN1, FGA, HIST1H2BK, ITIH2, KRT19, MARCH2, MAL2, and TIMP1) outperformed CA19.9 in discriminating PDAC from CP patients (AUC of 0.931 vs. 0.873). This poly-exo-mRNA signature could detect early stage PDAC (stage I and II) with an AUC = 0.949 and showed high consistency and robustness in the training (AUC = 0.960), internal validation (AUC = 0.950), and external validation (AUC = 0.936) cohorts [173]. 

Other studies that have investigated the diagnostic potential of different circulating and exoRNA single and panel molecules are summarised in Table 1. 

### 6.4. CTCs 

While most of the research to date has concentrated on RNAs, CA19.9, CEA, and their pure or mixed signatures, some studies on CTCs achieved a very high SP with a robust SN. Remarkably, three studies attained 100% SP [174,175,176], with one of them reaching 100% SN as well [176]. It should be kept in mind that study cohorts were small (*n* = 15–41 for PDAC and *n* = 15–20 for HC). A more recent publication involving a larger cohort (*n* = 100 for PDAC, *n* = 30 for HC, and *n* = 16 for IPMN) identified vimentin^+^ CTCs (v^+^ CTCs) in 76% of PDAC, 6.6% of HC, and 31% of IPMN samples [177]. Expectedly, v^+^ PDAC-derived CTCs exhibited a noticeably higher migratory capacity (than EpCAM positive ones), implying that vimentin could be used as a surface biomarker for PDAC-originating cells with a mesenchymal/basal phenotype. Additionally, v^+^ CTCs were mainly detected in patients in more advanced stages and with metastasis. The inclusion of CTCs (vimentin^+^, CD45^−^, and Hoechst^+^) improved the accuracy of CA19.9, achieving an AUC of 0.968 [177].

## 7. Prognostic and Predictive Potential of Liquid Biopsy Biomarkers

### 7.1. Protein Biomarkers

CA19.9 has been used not only as the diagnostic tool for PDAC, but also as a prognostic marker. Hence, it was shown that patients with resectable PDACs who had preoperative CA19.9 levels higher than the median had worse OS [184]. Likewise, another study showed lowered median preoperative CA19.9 values in N0 patients (without nodal disease) and an association of postoperative CA19.9 serum decrease (especially if falling under 200 U/mL) with improved OS [185]. Finally, it was demonstrated that patients receiving chemotherapy for locally advanced unresectable or metastatic PDAC who had CA19.9 values below 37 U/mL had a significantly longer OS compared to those with CA19.9 > 37 U/mL [186]. 

One of the key PDAC features is its dense fibrotic stroma. During development, the stroma is constantly rearranged in order to withstand the expansion of the tumour. That process is conducted not only by cancer cells, but with the aid of cancer-associated fibroblasts (CAFs) as well. Consequently, extracellular matrix (ECM) components of the stroma enter the bloodstream [187]. Hence, osteopontin (OPN), one of the ECM proteins, has been shown to be overexpressed in PDAC samples, and its elevated values (>150 ng/mL) were reported to be in correlation with worse OS [188]. In a more recent study, several ECM-related proteins were measured using ELISA in pre-treatment serum from stage III/IV PDAC patients who received 5-fluorouracil. Higher levels of all collagen fragments seemed to be associated with significantly shorter OS. Thus, elevated pro-peptides of type III collagen (PRO-C3, formation) correlated with worse OS, whereas a higher ratio of matrix metalloprotease-degraded type III collagen (C3M, degradation) and PRO-C3 (C3M/PRO-C3) led to better OS [189].

### 7.2. ctDNA

A study involving 105 PDAC patients who had undergone pancreatoduodenectomy compared OS and DFS according to ctDNA detection and presence [190]. Although mutKRAS (detected by ddPCR) was identified in 86 primary tumour samples, mutKRAS ctDNA was recognised in 31% of matching plasma specimens. Expectedly, patients who were ctDNA positive had worse OS and disease-free survival (DFS) [190]. In a small study with 27 patients, KRAS mutations were detected in 70.4% of individuals at baseline. Even though there was no statistically important difference in PFS and OS according to the presence or absence of mutKRAS ctDNA at the baseline, a clear correlation between PFS and the changes in mutKRAS ctDNA levels after the first cycle of chemotherapy was shown. Patients who had increased ctDNA levels at Day 15 had worse outcomes than those who had stable or decreased levels, and a two-month radiological evaluation confirmed the disease progression [191].

Wei et al. investigated the use of ctDNA as a biomarker for tumour burden and prognosis in PDAC patients who received FOLFIRINOX chemotherapy. They measured the ctDNA mutant allele frequency (MAF) in plasma samples from 28 patients and found that patients with multiple liver metastases had higher ctDNA MAF than those with one or two lesions, indicating a positive correlation between ctDNA MAF, and tumour burdens. They also divided the patients into two groups based on a ctDNA MAF cutoff of 1.5% and compared their OS. They observed that patients with high ctDNA MAF (≥1.5%) had worse OS than those with low ctDNA MAF (<1.5%). However, this result was not confirmed by multivariate Cox regression analysis, possibly due to the small sample size. Moreover, they reported that 12 patients who had a partial response or stable disease after chemotherapy showed a significant decrease in ctDNA MAF, except for one patient [192]. 

Other significant studies concerning prognostic and predictive potential of ctDNA are summarised in Table 2.

### 7.3. Exosomes, exoDNA, exoRNA, and Cell-Free microRNA Signatures

The role of exosomes and their surface molecules in predicting and influencing the outcomes of PDAC has been extensively studied. A study of Bernard et al. involved 194 PDAC patients with locally advanced or metastatic disease who received treatment. A subset of 34 locally advanced PDAC patients were followed longitudinally during neoadjuvant chemotherapy. A key finding was that a reduction in exo-mutKRAS MAF after neoadjuvant chemotherapy (compared to baseline) was significantly associated with surgical resection (which occurred in 12 out of 17 patients who underwent surgery). Patients who were not eligible for surgery (the remaining *n* = 17) had either an increase or no significant change in exo-mutKRAS MAF in 94% of the cases. Among metastatic patients, those who progressed or died during observation had worse survival with mutKRAS in both ctDNA and exoDNA. Therefore, while mutKRAS ctDNA had prognostic value, as discussed earlier, its exosomal counterpart seemed to perform better and provide earlier information [69]. 

Some publications have also highlighted the role of miRNAs and have shown that PDAC-derived exosomes carrying miR-222 can enhance cell invasion and proliferation, which can also induce increased cell survival and metastasis. Therefore, high levels of exo-miR-222 from plasma were significantly associated with tumour size and the TNM stage, and it became an independent risk factor for PDAC patients’ survival [196]. Exosomal miR-451a, when present in high amounts, has been shown to be an indicator of notably worse OS and DFS [197]. Additionally, elevated miR-451a combined with high expression of exo-miR-4525 and exo-miR-21 was significantly associated with recurrence [198]. In PDAC and IPMN patients, exo-miR-21, exo-miR-451a, and exo-miR-191 showed a significant increase in expression compared to the HC. This led to a correlation between exo-miR-451a and mural nodules in IPMN, a potential role of exo-miR-21 as a prognostic indicator for the OS, and their statistical significance as biomarkers of chemo-resistance [199]. Reese et al. demonstrated that miR-200b in EpCAM-positive serum exosomes could serve as an independent predictor of PDAC prognosis according to multivariate analysis [200]. 

Several studies have suggested that plasma miRNA profiles can be useful for predicting and assessing the outcomes of PDAC patients. For example, miR-744 was found to be highly expressed in PDAC tissues and cells compared to normal counterparts, and its plasma levels were associated with lymph node metastasis, recurrence, and poor PFS independent of other factors. Moreover, miR-744 overexpression conferred gemcitabine resistance in vitro [201]. Another study reported that plasma levels of miR-181a-5p were significantly reduced in patients who showed no progression after FOLFIRINOX treatment. This reduction was correlated with better OS and PFS, especially in patients who had a significant drop in CA19.9 as well. However, this combination of markers did not correlate with PFS and OS in patients who received gemcitabine–nab-paclitaxel treatment [202]. 

### 7.4. Circulating Tumour Cells

Possible ways to use CTCs to predict how well a treatment works are under active investigation for different kinds of cancer. Interestingly, an approach combining GPC1+ exosomes and CTCs to enhance the prognostic and/or predictive value was proposed several years ago. Neither of them was prognostic alone, but when combined, patients with >20% GPC1+ vesicles and/or CellSearch^®^ CTC+ clusters had worse OS. For PFS, poorer outcomes were linked to the presence of GPC1+ exosomes and/or CellSearch^®^ CTC+ clusters [203]. In a study including 41 samples that investigated the influence of the first round of chemotherapy on the number of detected CTCs, 80.5% of PDAC samples were reported positive for the CTC detection, while after administering 5-fluorouracil, only 29.3% of them were positive. The study also found that some apoptotic CTCs were detected after the first round of treatment in patients with advanced PDAC, which could be a potential indicator of the chemotherapy efficacy [174]. Meta-analysis conducted several years later confirmed that patients with detected CTCs had significantly worse OS and PFS than patients without CTCs [204]. 

Interestingly, a study aiming to evaluate the usefulness of real-time RT-PCR for EpCAM detection (used as a surrogate marker for the quantification of CTCs) reported no correlation between EpCAM positivity and DFS at any time point [116]. In a more recent comprehensive study that included 136 PDAC patients, 56 patients received neoadjuvant therapy prior to surgery and had significantly fewer CTCs compared to chemo-naïve patients. The preoperative numbers of all CTC subtypes were the only predictors of early recurrence within 12 months after resection in this cohort. Patients who received neoadjuvant chemotherapy and had no CTCs detected had significantly better OS, while in the chemo-naïve group, the association between CTCs and worse OS was not statistically significant. Moreover, they discovered that elevated levels of CTCs preceded recurrence [205]. Finally, another meta-analysis confirmed what many mentioned studies reported; CTC+ patients had significantly worse OS and PFS than CTC- patients [206]. 

## 8. Conclusions and Clinical Perspectives

PDAC is a deadly disease that requires early detection for effective treatment. Liquid biopsies are a promising technique that can provide noninvasive, repeatable, and real-time screening of PDAC patients. However, PDAC is a complex and heterogeneous disease that may share molecular features with other cancers and other pathological conditions, which can affect the accuracy and reliability of potential biomarkers. 

Therefore, a better strategy may be to look for a combination of multiple biomarkers that can distinguish PDAC from other conditions. A novel direction may be to investigate the role of intercellular communication, especially through exosomes and their cargo, in PDAC progression and diagnosis. Taking into account that exoRNAs stem from living cells and that ctDNAs are the consequence of apoptosis and/or necrosis, it has been proposed to combine them in order to account for tumoural heterogeneity [94]. This area has a lot of potential for improvement, especially in identifying the surface proteins of exosomes that originate from PDAC cells. 

None of the components of liquid biopsies apart from CA19.9 have become part of PDAC clinical practice so far. For some, this is due to unsatisfactory SN and/or SP values and the shortfall of their validation, for others it is because of a lack of standardization and automatization of appropriate lab methods. However, liquid biopsies are a prospective tool in PDAC patient care and follow-up. It would be of invaluable help and importance for both patients and the doctors to reach the stage of PDAC clinical practice where liquid biopsies could be used for observation of the treatment efficacy, unhesitating prognosis, and assessment of a relapse simply by testing the presence and level of one or several blood-derived molecules. Interestingly, they might be able to indicate the direction and route that the disease has taken. Another invaluable application for potential biomarkers could be to act as navigators for resectable tumours, hence facilitating decision-making between neoadjuvant therapy and immediate surgery. Moreover, detection of a certain marker may represent a recent metastasis onset, thus sparing patients a futile resection.

## Figures and Tables

**Figure 1 cells-13-00003-f001:**
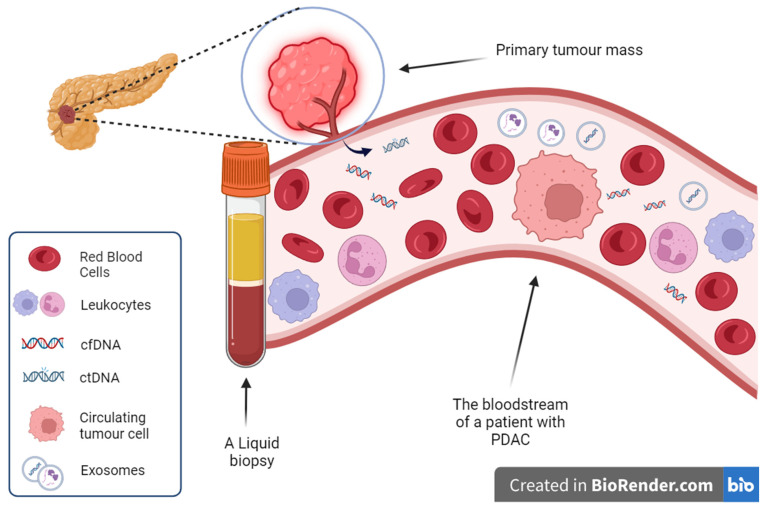
Cell-free DNA, circulating tumour DNA, and other blood components of an oncology patient.

**Figure 2 cells-13-00003-f002:**
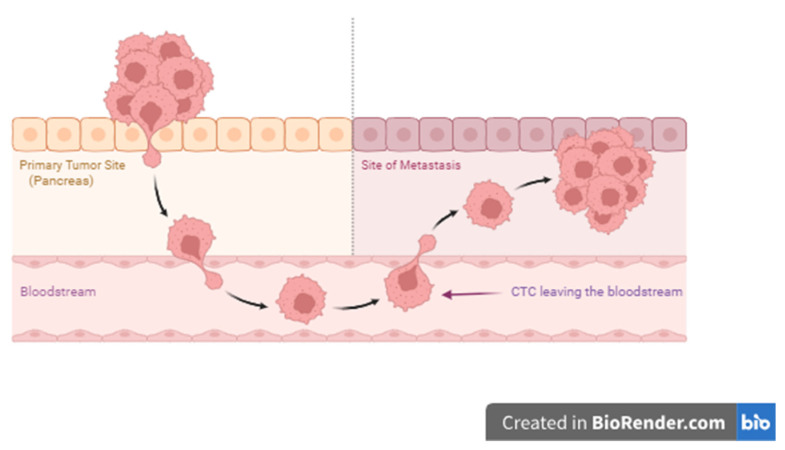
Circulating tumour cells.

**Table 1 cells-13-00003-t001:** **Discrimination and diagnostic potential of various biomarkers of PDAC**. Listed biomarkers were curated according to their sensitivity (SN), specificity (SP), and area under the ROC curve (AUC) values. NM (not mentioned).

Biomarker	Patients	SN (%)	SP (%)	AUC	Reference	Year
PROTEIN BIOMARKERS						
CA19.9	3285 PDAC vs. 1882 cases with benign pancreatic disease	78.2	82.8	Differs for malignant vs. benign cases (0.878) and PDAC vs. CP (0.885)	[30]	2013
CEA	1324 PDAC vs. 301 cases with benign pancreatic disease	44.2	84.8	Differs for malignant vs. benign cases (0.702) and PDAC vs. CP (0.721)	[30]	2013
ExoGPC-1	246 PDAC vs. 120 HC	100	100	1.0	[25]	2015
HSP-27	35 PDAC vs. 37 HC	100	84	0.98	[129]	2007
COL6A3	44 PDAC vs. 30 HC	93	97	0.975	[59]	2014
CXCL8	42 PDAC vs. 34 HC	98	95	0.9898	[130]	2018
REG1A and REG1B	41 PDAC vs. 61 HC	92	95	NM	[131]	2016
PIM-1	90 PDAC vs. 20 HC	95.6	100	0.984	[132]	2016
MIC-1	2770 PDAC vs. 2082 HC	NM	NM	0.93	[133]	2023
PROTEIN PANELS						
TFPI, TNC, and CA19.9	37 PDAC vs. 15 HC	90	100	0.99	[134]	2011
ICAM-1, OPG, and CA19.9	333 PDAC vs. 227 HC	88	90	0.93	[135]	2011
C5, A1BG, and CA19.9	22 PDAC vs. 29 HC	87	90	0.92	[136]	2013
C4BPA and CA19.9	52 PDAC vs. 40 HC	85	96	0.93	[137]	2016
IGFBP2, IGFBP3, and CA19.9	101 PDAC vs. 38 HC	88	89	0.89	[138]	2016
THBS2 and CA19.9	288 PDAC vs. 230 HC	87	87	0.97	[139]	2017
TIMP1, LRG1, and CA19.9	187 PDAC vs. 169 HC	85	95	0.95	[140]	2017
ALB, CRP, IL-8, and CA19.9	292 PDAC vs. 383 HC	94	90	0.98	[141]	2014
APOA2-ATQ/AT and CA19.9	286 PDAC vs. 217 HC	95.4	98.3	0.96	[142]	2015
APOA2, APOC1, and CA19.9	111 PDAC vs. 105 HC	93	100	0.96	[143]	2010
APOA1, APOE, APOL1, ITIH3, and CA19.9	80 PDAC vs. 40 HC	95	94.1	0.99	[144]	2017
APOA1, APOE, APOL1, and ITIH3	80 PDAC vs. 40 HC	85	94	0.94	[144]	2017
CA242, CA19.9, CEA, and CA125	52 PDAC vs. 40 HC	90	94	NM	[145]	2015
POSTN, CA242, and CA19.9	213 PDAC vs. 74 HC	92	97	0.98	[146]	2018
EPHB3, FGF1, ID1, IL2, IL10, IMPDH2, SELL, and VCAM1	72 PDAC vs. 49 HC	89	91	0.95	[147]	2017
10 peptide signatures	88 PDAC vs. 185 HC	92% and 95%	95	0.96	[148]	2015
HPT, C3, C4A, C5, C7, IgG1, and IgA1	122 PDAC vs. 252 HC	92.1	90.6	0.94	[149]	2014
18 proteins targeted by scFv human recombinant antibodies	103 PDAC vs. 30 HC	88	85	0.95	[150]	2012
19 proteins targeted by scFv human recombinant antibodies	156 PDAC vs. 30 HC	99	80	0.98	[151]	2015
29 proteins targeted by scFv human recombinant antibodies	586 PDAC vs. 1107 HC	95	94	0.97	[152]	2018
CIRCULATING TUMOUR DNA						
Quantity of cfDNA	24 PDAC vs. 38 HC and 21 IPMN vs. 38 HC	83 and 81	92 and 84	0.92	[153]	2016
DNA methylation of SST	30 PDAC vs. 18 HC	93	89	0.89	[154]	2020
DNA methylation of ADAMTS1 and BNC1	42 PDAC vs. 26 HC	81	85	NM	[155]	2013
DNA methylation of ADAMTS1 and BNC1	39 PDAC vs. 95 HC	97	92	0.95	[156]	2019
Mutations in amplicons	100 PDAC vs. 29 HC	82	100	NM	[157]	2016
RNA BIOMARKERS						
Exo-miRNA-21	30 PDAC vs. 10 CP	80	90	NM	[158]	2020
Exo-miRNA-21	22 PDAC vs. 27 non-PDAC	NM	NM	0.897	[159]	2013
miR-18a	36 PDAC vs. 30 HC	92	94	0.9369	[160]	2011
miR-1290	19 PDAC vs. 10 HC, 19 PDAC vs. 10 CP and 19 PDAC vs. 10 NPET	88 for PDAC vs. HC	84 for PDAC vs. HC	0.96, 0.81 and 0.80	[62]	2013
miR-22-3p	35 PDAC vs. 15 HC	97.14	93.33	0.943	[161]	2017
miR-642b-3p	35 PDAC vs. 15 HC	100	100	1.0	[161]	2017
miR-885-5p	35 PDAC vs. 15 HC	100	100	1.0	[161]	2017
Exo-miR-21	27 PDAC vs. 8 CP	81	88	0.89	[162]	2019
Exo-miR-155	27 PDAC vs. 8 CP	89	88	0.90	[162]	2019
Exo-miR-451	52 PDAC vs. 20 HC	NM	NM	0.9329	[163]	2021
Exo-miR-720	52 PDAC vs. 20 HC	NM	NM	1.0	[163]	2021
miR-373	103 PDAC vs. 50 HC	81	84	0.852	[164]	2017
WASF2	27 PDAC vs. 13 HC	NM	NM	0.943	[83]	2019
ARF6	27 PDAC vs. 13 HC	NM	NM	0.940	[83]	2019
SNORA74A	27 PDAC vs. 13 HC	NM	NM	0.909	[83]	2019
SNORA25	27 PDAC vs. 13 HC	NM	NM	0.903	[83]	2019
HULC	20 PDAC vs. 21 HC and 20 PDAC vs. 22 IPMN	80 and 85	95 and 83	0.94 and 0.91	[165]	2020
MIXED AND RNA PANELS						
Exo-miR-10b, 21, 30c, 181a, and let7a	29 PDAC vs. 6 HC and 29 PDAC vs. 11 CP	100	100	1.0	[63]	2017
miR-16, miR-196a, and CA19.9	140 PDAC vs. 68 HC and 140 PDAC vs. 111 CP	92 and 88.4	95.6 and 96.3	0.979 and 0.956	[166]	2012
miR-1290, miR-1246, and CA19.9	120 PDAC vs. 40 HC and 120 PDAC vs. 40 Non-PDAC (CP/IPMN/PNET)	96.7 and 92.5	97.5 and 90	0.99 and 0.96	[167]	2020
miR-125a, miR-4294, miR-4476, miR-4530, miR-6075, miR-6799, miR-6836, and miR-6880	100 PDAC vs. 150 HC	80.3	97.6	0.953	[168]	2015
miR-125a-3p, miR-642b-3p, and miR-5100	424 PDAC vs. 2599 HC	98	97	0.95	[169]	2020
Signature of 10 miRNAs	409 PDAC vs. 312 HC	85	85	0.93	[64]	2014
Signature of 12 miRNAs	417 PDAC vs. 307 HC	85	90	0.95	[170]	2016
LGLRAD3 and CA19.9	31 PDAC vs. 31 HC	80	94	0.87	[171]	2017
ABHD11-AS1 and CA19.9	114 PDAC vs. 46 HC	98	100	0.98	[172]	2019
Exo-CLDN1, FGA, HIST1H2BK, ITIH2, KRT19, MARCH2, MAL2, and TIMP1	189 PDAC vs. 74 HC and 186 PDAC vs. 55 CP	96 and 94	100 and 81	0.98 and 0.92	[173]	2020
CIRCULATING TUMOUR CELLS						
CK8, CK18, and CA19.9	41 PDAC vs. 20 HC	80	100	NM	[174]	2011
CD45^−^, CK8, CK18, and CK19	15 PDAC vs. 15 HC	80	100	NM	[175]	2015
Expression of C-MET, hTERT, CK20, and CEA	25 PDAC vs. 15 HC	100	100	NM	[176]	2011
Vimentin^+^, CD45^−^, Hoechst^+^, and CA19.9	100 PDAC vs. 30 HC	91	97	0.97	[177]	2019

**Table 2 cells-13-00003-t002:** **Prognostic and predictive potential of various biomarkers.** Listed biomarkers are curated according to their overall survival (OS).

Biomarker	OS	Reference	Year
PROTEIN BIOMARKERS			
CA19.9 < 37 U/mL	Better prognosis	[186]	2013
CA19.9 > 37 U/mL	Worse prognosis	[186]	2013
OPN < 150 ng/mL	337 days vs. 179 days	[188]	2013
Elevated PRO-C3	Worse prognosis	[189]	2019
Elevated ratio C3M/PRO-C3	Better prognosis	[189]	2019
CIRCULATING TUMOUR DNA			
MutKRAS (G12D, G12V, and G12R)	13.6 vs. 27.6 months	[190]	2016
MutKRAS (G12V)	4.7 vs. 6.0 months	[193]	2017
ERBB2 exon 17 mutation	4.7 vs. 5.7 months	[193]	2017
MutKRAS (G12D)	6.5 vs. 11.5 months	[191]	2017
CtDNA MAF ≥ 1.5%	Worse prognosis	[192]	2019
MutKRAS	Worse prognosis	[194]	2018
MutKRAS	Worse prognosis	[195]	2019
EXO-DNA, EXO-RNA, AND CELL-FREE MICRO RNA			
MAFs ≥ 5% in exoDNA	Worse prognosis	[69]	2019
Exo-miR-222	10 vs. 17 months	[196]	2018
Exo-miR-451a	Worse prognosis	[197]	2018
Exo-miR-4525, exo-miR-451a, and miR-21	Worse prognosis	[198]	2019
Exo-miR-21	344 vs. 846 days	[199]	2018
Exo-miR-200b in EpCAM positive exosomes	9 vs. 18 months	[200]	2020
Exo-miR-200c in total serum exosomes	11 vs. 18 months	[200]	2020
Cell free miR-744	Not mentioned	[201]	2015
Combination of miR-181a-5p and CA19.9 (only in patients receiving FOLFIRINOX)	11.1 vs. 25.7 months	[202]	2020
GPC1+ exosomes	Worse prognosis	[25]	2015
CIRCULATING TUMOUR CELLS			
>20% GPC1+ vesicles and/or CellSearch^®^ CTC+ clusters	Worse prognosis	[203]	2019
CTC+	Worse prognosis	[204]	2014
CTC+ in patients who received neoadjuvant chemotherapy	Worse prognosis	[205]	2018
CTC+	Worse prognosis	[206]	2020

## Data Availability

Not applicable.

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
