# Peer review of "A Comprehensive Review of the Potential Role of Liquid Biopsy as a Diagnostic, Prognostic, and Predictive Biomarker in Pancreatic Ductal Adenocarcinoma"

_cells, 2023, doi:10.3390/cells13010003_

Round 1
Reviewer 1 Report
Comments and Suggestions for Authors
I just want to congratulate the authors of the manuscript. They have included all relevant studies regarding pancreatic cancer and liquid biopsy. They are accurate in their conclusions.
If I had something to mention, that would have been to add a couple of phrases in the "introduction" regarding liquid biopsy in pancreatic cancer such as history or when was the first attempt or the first study or publication.
Author Response
We want to take the opportunity to thank reviewer 1 for his review report and suggestions. The introduction was revised accordingly to include the historical aspect and first works related to liquid biopsy analyses in pancreatic cancer (lines 76-90 in track change version).
Reviewer 2 Report
Comments and Suggestions for Authors
Authors reviewed recent findings on liquid biopsy usefulness for management of PDAC. The idea of paper is quite interesting, however, paper has some limitations that have to be corrected:
1. Introduction can be shorter and more concrete
2. The protein biomarkers are well known, and they are not exact biomarkers in the mention - rather used for monitoring or disease recurrence. I suggest to only mention a bit about such biomarkers or compare them with molecular ones. More interesting will be to focus on molecular biomarkers detectable with the use of liquid biopsy
3. The subchapters with isolation methods for cfDNA and exRNAs (also the figure 2, it is alreadu obvious) are unnecessary, of course authors can mention about them but briefly. Instead of this subchapters, it will be better to discuss liquid biopsy as well as pros and cons of this technique in clinical setting.
4. The chapter on CTCs should rather focus on molecular alterations, such as mutations or methylation and they role in clinics.
5. Table 1 is too voluminious, and it is hard to read. I suggest to divide table according to groups of molecular alteranions. Alternatively, some ot these information can appear in the new figure.
6. Table 2 consists of numerous NM informations. To be more convenient, the studies containing most of that data should be mentioned only in the text.
Author Response
Response to reviewer 2
We want to take the opportunity to thank the reviewer for the constructive comments and suggestions to improve the manuscript. The manuscript was revised accordingly to take into account the comments of the reviewer 2. More precisely:
- Introduction can be shorter and more concrete
Response :
A couple of paragraphs in the introduction have been shortened (lines 40-44 ; 61-62 ;: 64-65)) and we believe that the additional text added in lines 76-90 based on the suggestion of the reviewer 1 is making the introduction clearer and more to the point.
- The protein biomarkers are well known, and they are not exact biomarkers in the mention - rather used for monitoring or disease recurrence. I suggest to only mention a bit about such biomarkers or compare them with molecular ones. More interesting will be to focus on molecular biomarkers detectable with the use of liquid biopsy
Response
We agree with reviewer 2 that CA19-9 and CEA are well known and mainly used for monitoring and disease recurrence, but as they are currently the only blood-based biomarkers in clinical use for PDAC , we considered worth describing them very briefly in chapter 2, even more so as they are incluided in the biomarker panels and mentioned many times in the manuscript later on.
- The subchapters with isolation methods for cfDNA and exRNAs (also the figure 2, it is alreadu obvious) are unnecessary, of course authors can mention about them but briefly. Instead of this subchapters, it will be better to discuss liquid biopsy as well as pros and cons of this technique in clinical setting.
Response :
As per reviewer 2 suggestion, we deleted the Figure 2. As the selection of the adequate isolation method has been proven instrumental for further downstream liquid biopsy analyses, we shortened, rather than deleted entirely, the cfDNA and exosomes isolation methods.
Considering the the pros and cons of the techniques to be used in clinical setting, they have been adressed in chapters dedicated to each liquid biopsy component.
- The chapter on CTCs should rather focus on molecular alterations, such as mutations or methylation and they role in clinics.
Response :
The chapter on CTC has been reviewed to include molecular alterations ( lines 310-325 in track changes version).
- Table 1 is too voluminious, and it is hard to read. I suggest to divide table according to groups of molecular alteranions. Alternatively, some ot these information can appear in the new figure.
Response :
Table 1 has been separated with subheaders according to the groups of molecular alterations.
- Table 2 consists of numerous NM informations. To be more convenient, the studies containing most of that data should be mentioned only in the text.
Response:
Table 2 was reformatted similarly to the Table 1 (subheaders were added for more clarity) and columns containing some ‘NM’ informations were deleted from the table and the studies containing most of the data were mentioned in the text.
Reviewer 3 Report
Comments and Suggestions for Authors
This article is a literature review highlighting advances in the detection of new markers in liquid biopsy for pancreatic cancer, and the value of these biomarkers in disease early detection, as a prognostic factor and as an indicator of response to systemic treatment.
This review is exhaustive, with well-referenced data. it provides a good understanding of the state of knowledge on the subject of tumor biomarkers in pancreatic cancer. The techniques for isolating, enriching and interpreting these biomarkers are well explained, and the article's conclusions are consensual with the rest of the literature. Indeed, the use of combined biomarkers and the presence of a biomarker signature are more relevant and useful for clinical purposes than alone, considering PDAC heterogeneity.
References are very exhaustive and recent, tables provide a comprehensive overview of the sensitivity and specificity of the various biomarkers tested and their prognostic value, alone and combined. Figures are clear and relevant.
This is an interesting and relevant article on the state-of-the-art of biomarkers in liquid biopsy for PDAC, despite several similar articlesThis article is a literature review highlighting advances in the detection of new markers in liquid biopsy for pancreatic cancer, and the value of these biomarkers in disease early detection, as a prognostic factor and as an indicator of response to systemic treatment.
This review is exhaustive, with well-referenced data. it provides a good understanding of the state of knowledge on the subject of tumor biomarkers in pancreatic cancer. The techniques for isolating, enriching and interpreting these biomarkers are well explained, and the article's conclusions are consensual with the rest of the literature. Indeed, the use of combined biomarkers and the presence of a biomarker signature are more relevant and useful for clinical purposes than alone, considering PDAC heterogeneity.
References are very exhaustive and recent, tables provide a comprehensive overview of the sensitivity and specificity of the various biomarkers tested and their prognostic value, alone and combined. Figures are clear and relevant.
This is an interesting and relevant article on the state-of-the-art of biomarkers in liquid biopsy for PDAC, despite several similar articles
Author Response
We thank the reviewer 3 for investing time to review our manuscript and his encouraging review report.
Reviewer 4 Report
Comments and Suggestions for Authors
The Authors presented a very well structured Review regarding the use of liquid biopsy in the pathology of PDAC. The figures presented represent an original point of view of the writers. The list of biomarkers also seems very large and comprehensive. The number of references is adequate. The present work constitutes a valid tool to summarize the usefulness of liquid biopsy for investigating the pathology of PDAC.
Comments on the Quality of English LanguageThe review presented by the Authors is suitable for publication the present form.
Author Response
We thank the reviewer 4 for investing time to review our manuscript and his encouraging review report.
Round 2
Reviewer 2 Report
Comments and Suggestions for Authors
Authors corrected their paper accordingly. I have no further comments.